# Media and social media attention to retracted articles according to Altmetric

**Stylianos Serghiou**[1,2], **Rebecca M. Marton**[2], **John P. A. Ioannidis**[1,2,3,4]*

**1** Department of Epidemiology and Population Health, Stanford University School of Medicine, Stanford, CA, United States of America, **2** Meta-Research Innovation Center at Stanford (METRICS), Stanford School of Medicine, Stanford, CA, United States of America, **3** Department of Medicine, Stanford Prevention Research Center, Stanford University School of Medicine, Stanford, CA, United States of America, **4** Department of Statistics, Stanford University School of Humanities and Sciences, Stanford, CA, United States of America

* jioannid@stanford.edu

## Abstract

The number of retracted articles has grown fast. However, the extent to which researchers and the public are made adequately aware of these retractions and how the media and social media respond to them remains unknown. Here, we aimed to evaluate the media and social media attention received by retracted articles and assess also the attention they receive post-retraction versus pre-retraction. We downloaded all records of retracted literature maintained by the Retraction Watch Database and originally published between January 1, 2010 to December 31, 2015. For all 3,008 retracted articles with a separate DOI for the original and its retraction, we downloaded the respective Altmetric Attention Score (AAS) (from Altmetric) and citation count (from Crossref), for the original article and its retraction notice on June 6, 2018. We also compared the AAS of a random sample of 572 retracted full journal articles available on PubMed to that of unretracted full articles matched from the same issue and journal. 1,687 (56.1%) of retracted research articles received some amount of Altmetric attention, and 165 (5.5%) were even considered popular (AAS>20). 31 (1.0%) of 2,953 with a record on Crossref received >100 citations by June 6, 2018. Popular articles received substantially more attention than their retraction, even after adjusting for attention received post-retraction (Median difference, 29; 95% CI, 17–61). Unreliable results were the most frequent reason for retraction of popular articles (32; 19%), while fake peer review was the most common reason (421; 15%) for the retraction of other articles. In comparison to matched articles, retracted articles tended to receive more Altmetric attention (23/31 matched groups; P-value, 0.01), even after adjusting for attention received post-retraction. Our findings reveal that retracted articles may receive high attention from media and social media and that for popular articles, pre-retraction attention far outweighs post-retraction attention.

## Introduction

Retraction refers to the formal withdrawal of a publication, most often due to scientific misconduct or an error that invalidates the purported conclusions [1]. The number of retracted

**Data Availability Statement:** Data extracted from Altmetric, CrossRef and all matched PubMed articles have been deposited on OSF (Open Science Framework) and may be accessed at https://www.doi.org/10.17605/OSF.IO/7T32U under a CC-By

**Funding:** The authors received no specific funding for this work. METRICS (J.P.A.I) has been supported by grants from the Laura and John Arnold Foundation. S.S. has been funded by the Department of Epidemiology and Population Health at Stanford University and as a Scholar of the Stanford Data Science Initiative. The funders had no role in study design, data collection and analysis, decision to publish, or preparation of the manuscript.

**Competing interests:** The authors have declared that no competing interests exist.

articles has increased dramatically over the last decade, with less than 100 reported per annum before 2000, to almost 1000 in 2014 [2] and 1,772 in 2019 [3]. Such retractions are often publicised by the journal itself in the form of a retraction notice (albeit not all journals issue a retraction notice upon retraction) and initiatives, such as Retraction Watch [4] of the Center for Scientific Integrity, keep track of these retractions. However, the extent to which researchers or the public are made aware of these retractions and the amount of attention that they receive is unknown.

Concerningly, current evidence suggests widespread misinformation. As of May 2019, the most highly cited retracted article had received 371 citations since its retraction in 2018 and seven out of ten most highly cited retracted articles had received at least 100 citations since their retraction [5]. Some of these citations may be citing the work as being unreliable and acknowledging its retraction, but this is not necessarily the case in many citations. For example, in a study of all 25 papers by anaesthesiologist Scott S. Reuben that have been retracted, it transpired that 74% of citations received post-retraction did not clearly state that the work they were referring to had been retracted [6]. These results align well with literature on case studies [7, 8], specific disciplines [9] and the broader literature [10], which identify that at least 80% of retracted articles receive positive post-retraction citations. Such perpetuated misinformation is not inconsequential: guidelines and meta-analyses seem to be very rarely updated to remove retracted articles [11] and a recent preprint suggests that doing so would lead to a median reduction in estimated effect size of 13% and an average reduction of 30% [12].

One particular type of impact for scientific articles is the attention they receive in media and social media. This type of impact is complementary to citations in the scientific literature and may be more relevant when it comes to understanding how an article fares in the wider community, beyond just expert scientists. It would be very interesting to understand how much attention retracted articles receive and how retraction may affect the attention that they receive. This is feasible to examine and to compare also against citation counts by using readily available databases. The Altmetric database [13] tracks any media or social media attention to articles with a digital object identifier (DOI) and Crossref [14] maintains a citation count for such articles.

We integrated the Retraction Watch Database [3], which systematically captures retracted articles, with data from Altmetric and Crossref to (a) describe retracted article characteristics and associated amount of impact and attention received, (b) compare whether the amount of attention received changed before and after retraction, (c) describe the amount of attention received by retracted articles in comparison to the amount received by their retraction notice, and (d) compare the amount of attention that eventually retracted articles received to that of similar matched unretracted articles published in the same journal issues.

## Results

### The RetractionWatch database

As of August 14, 2020, the RetractionWatch database contained 22,200 publications with a unique Digital Object Identifier (DOI), PubMed ID (PMID) or title (when DOI or PMID were not available) published between 1923 and 2020. Of these, we retained 11,807 unique publications published between 2010–2015 (S1 Table), which we chose *a priori* as a representative sample with sufficient time to accrue retractions and data about the impact of those retractions. Of these, most studies were designated by Retraction Watch as either conference abstracts (6,561; 56%), research articles (4,046; 34%) or clinical studies (450; 3.8%); overall, we identified 4,603 (39%) studies that we define as research articles (see Materials and Methods).

Most research articles were classified by Retraction Watch under at least one of Biological sciences (2,387; 52%), followed by the Health sciences (2,031; 44%) and the Physical sciences (1,233; 27%) (Table 1; S2 Table). The most common subcategory was cellular biology (1,060; 23%). There was a very large number of journals represented (n = 2,239 journals) and out of 392 publishers, the most commonly occurring were Elsevier (939; 20%), followed by Springer (719; 16%) and Wiley (333; 7.2%). The most highly represented countries were China (1,260; 27%), United States (891; 19%) and India (402; 8.7%) (Fig 1).

Out of 4,142 retraction notices with a unique DOI or PMID, 44 referred to the retraction of more than one original article—the largest retraction was published in Tumor Biology, retracting 103 unique articles due to fake peer review [15]. The commonest reasons for retraction were Duplication of article (667; 15%), Fake peer-review (594; 13%) and Plagiarism of article (412; 9%) (S3 Table). For all 4,609 unique article-retraction pairs, the median time from publication to retraction was 457 days (IQR, 179–956 days) (S1 Fig).

## Altmetric attention score and citations

Of 4,324/4,603 original articles with a DOI, 3,363 (81%) had a different DOI for the original article than the retraction notice. For 3,097 (92%) of these we extracted data about Altmetric attention and citations; the discrepancy exists because we are reporting on an updated version of the Retraction Watch Database than the one originally used to extract AAS and citation counts in June 6, 2018. Within these, the median Altmetric Attention Score (AAS) for an original article was 0.50 (Interquartile range (IQR), 0.00–7.3) and for a retraction notice it was 0.25 (IQR, 0.0–9.0) (Fig 2); the AAS is a composite measure of total media (e.g. news outlets) or social media (e.g. Twitter) attention (see Materials and Methods). Out of 3,097 research articles, 1,733 (56%) articles in our dataset received any media and social media attention (AAS>0) and 168 (5.4%) received substantial media and social media attention (AAS>20). These popular articles were published in 108 different journals, the most common being Science (12/168; 7%) and Nature (10, 6%) (Fig 3A). The publisher with most popular retracted articles was Springer-Nature (26/168; 15%) (Fig 3A). Articles with AAS<20 were published in a much larger array of journals (n = 1,445). The publisher with the most such retractions was Elsevier (542/2,923; 19%). The commonest reason for retraction of popular articles was Unreliable results (33/168; 20%) while for other articles it was Fake peer review (487/2,923; 17%).

Of 3,570 unique articles with Crossref citation data for both the original article and its retraction notice, 3,008 (84%) had a separate DOI for the original article and its retraction notice. The median citations for these 3,008 articles were 4 for original articles (IQR, 1–12) and 0 for retraction notices (IQR, 0–0) as of June 6, 2018 (Fig 2). 28 of the original articles (0.9%), but none of the retraction notices, received at least 100 citations. The most common journal for highly cited (>100 citations) retracted articles was Cell (both 4/28; 14%) (Fig 3B). The commonest reason for retraction of highly cited articles was Manipulation/Duplication of Images (8/28; 29%), whereas for other articles (<100 citations) it was Fake peer review (487/2,979; 16%).

## Attention and citations to the original article vs. its retraction notice

Overall, for 3,097 original articles and their retraction notice, the AAS of each original article did not differ substantially from its retraction notice (Median difference, 0; IQR, -1.0–1.0; P-value, 0.54). However, popular original articles received substantially higher media and social media attention than their retraction notice (Median difference, 30; IQR, 14–91; P-value < $10^{-16}$) (Fig 4). 109/168 (65%) popular articles did not have a popular retraction notice and 10/168 (6%) popular articles had a retraction notice with no attention received at all (AAS = 0).

**Table 1. Descriptive statistics for 4,603 unique eligible research articles.**

| | | Count | Percent |
|---|---|---|---|
| **Date (original)** | **2010** | 650 | 14% |
| | **2011** | 674 | 15% |
| | **2012** | 827 | 18% |
| | **2013** | 703 | 15% |
| | **2014** | 912 | 20% |
| | **2015** | 837 | 18% |
| **Date (retraction)** | **2010** | 128 | 3% |
| | **2011** | 301 | 7% |
| | **2012** | 473 | 10% |
| | **2013** | 576 | 12% |
| | **2014** | 619 | 13% |
| | **2015** | 949 | 21% |
| | **2016** | 860 | 19% |
| | **2017** | 526 | 11% |
| | **2018** | 175 | 4% |
| **Article type** | **Research Article** | 4,044 | 88% |
| | **Clinical Study** | 450 | 10% |
| | **Meta-analysis** | 120 | 3% |
| **Country** | **China** | 1,260 | 27% |
| | **United States** | 915 | 19% |
| | **India** | 402 | 9% |
| | **Iran** | 309 | 7% |
| | **South Korea** | 227 | 5% |
| | **Other (n = 110)** | 2,047 | 45% |
| **Retraction reason** | **Duplication of Article** | 667 | 15% |
| | **Fake Peer Review** | 594 | 13% |
| | **Plagiarism of article** | 412 | 9% |
| | **Other (n = 88)** | 3,737 | 81% |
| | | **Median** | **IQR** |
| **Date** | **Original** | 2013 | 2011–2014 |
| | **Retraction** | 2015 | 2013–2016 |
| **AAS** | **Original** | 0.25 | 0.00–7.00 |
| | **Retraction** | 0.25 | 0.00–8.79 |
| **Citation count** | **Original** | 3 | 1–10 |
| | **Retraction** | 0 | 0–0 |

For Article type, Country and Retraction reason the proportions do not add up to 100% because each article could be classified under multiple article types, have multiple reasons for retraction and have affiliations from multiple countries. Missing values: Country (1, 0%), AAS—Original (758, 17%), AAS—Retraction (847, 18%), Citation count—Original (817, 18%) and Citation count—Retraction (914, 20%). The large number of missing AAS and citation counts is due to the subsequent addition of articles not initially present in our version of the Retraction Watch Database (see Materials and Methods). IQR = Interquartile Range. AAS = Altmetric Attention Score.

Overall, the original article received more attention than its retraction notice on 1,056 occasions and the retraction notice more than the original article on 1,016 occasions (P-value, 0.39). For popular articles, the numbers were 156 versus 12 (P-value $< 10^{-16}$).

However, the above results do not take into account attention received by the original article because of its retraction. Within our sample of 3,097 articles, 279 were retracted within a

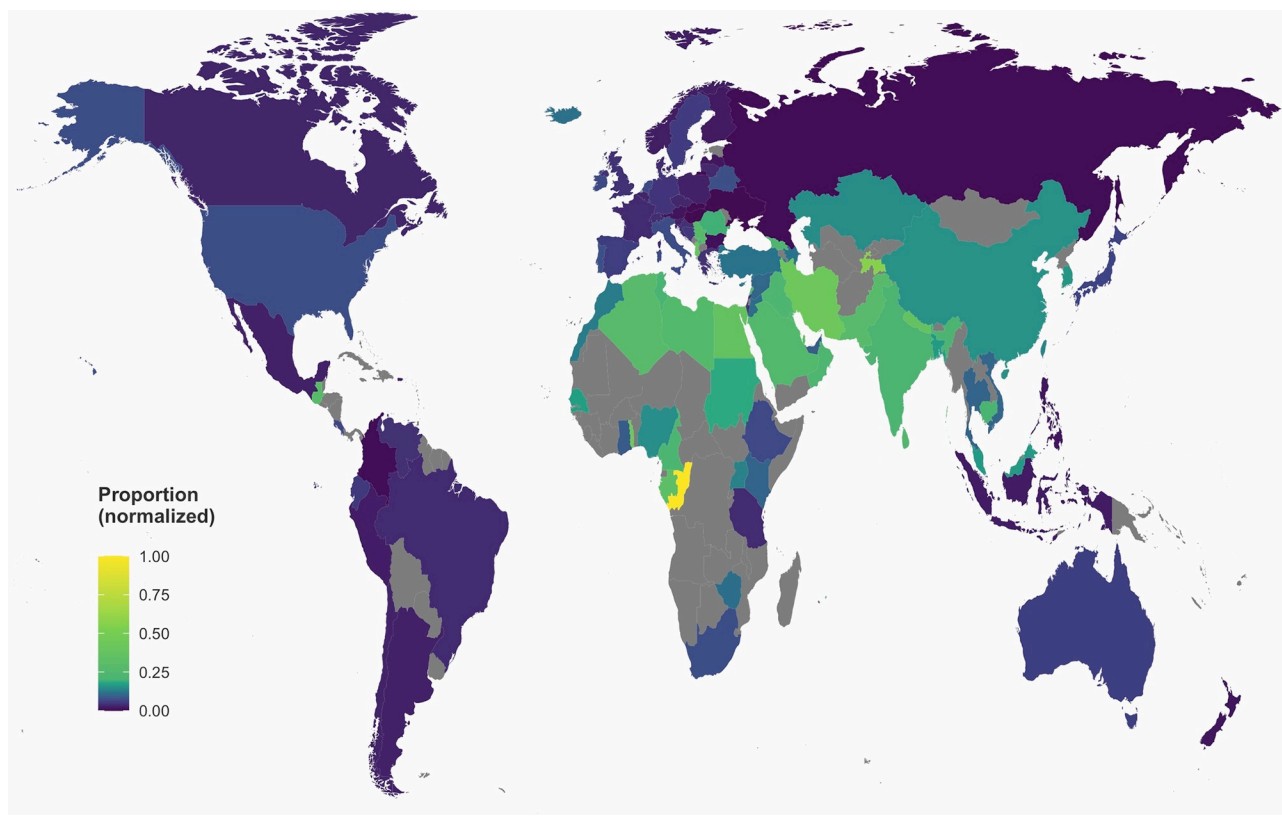

**Fig 1. Proportion of retracted research articles by country.** Proportion of retracted research in relation to all peer-reviewed documents published in 2010–2015 for countries with >500 peer-reviewed documents within those 6 years, as indicated by the National Science Foundation (see Materials and Methods). This varies substantially by country and by continent. The continent with most retractions is Asia and the continent with least retractions is Europe. The country with the highest proportion of retractions was the Republic of Congo (3/922; 0.3%) and the country with the least proportion was Hungary (3/55,609, 0.005%). Note that NSF counts of total peer-reviewed documents do not include letters, which represent 21/4,603 (0.5%) of our research articles as per our eligibility criteria. Grey signifies no data for those countries.

year since we retrieved data from Altmetric. For these articles we could use data provided by Altmetric on cumulative attention received across the past 1 month, 3 months, 6 months and 1 year. Over the previous year, the 279 articles gained most attention in the months following their retraction (Fig 5A), in contrast to articles retracted more than a year ago, which did not experience appreciable attention gain during the previous year (Fig 5B, S2 Fig). Of 279 articles retracted within a year of our data collection, 179 (64%) articles received at least some attention before or after retraction—87 (49%) received most attention before retraction and 88 (49%) before retraction (4 had similar attention before and after retraction).

Considering only the 279 articles for which we could retrieve data on changes in AAS over time, the effects observed when considering total AAS were attenuated (Table 2, S3 Fig). However, the median attention received by popular articles was still markedly higher than that of their retraction (Median difference, 31; IQR, 21–82; Median original-to-retraction ratio, 4.1; IQR of original-to-retraction ratio, 2.8–16.2). In considering 179 articles with non-zero original attention, the attention received by the original article exceeded that of the retraction most of the time (121 vs. 42), even though the median difference was small (Median, 0.8). These numbers were only slightly attenuated in sensitivity analyses where all of the attention received by the original over the last year was removed (instead of only removing attention received

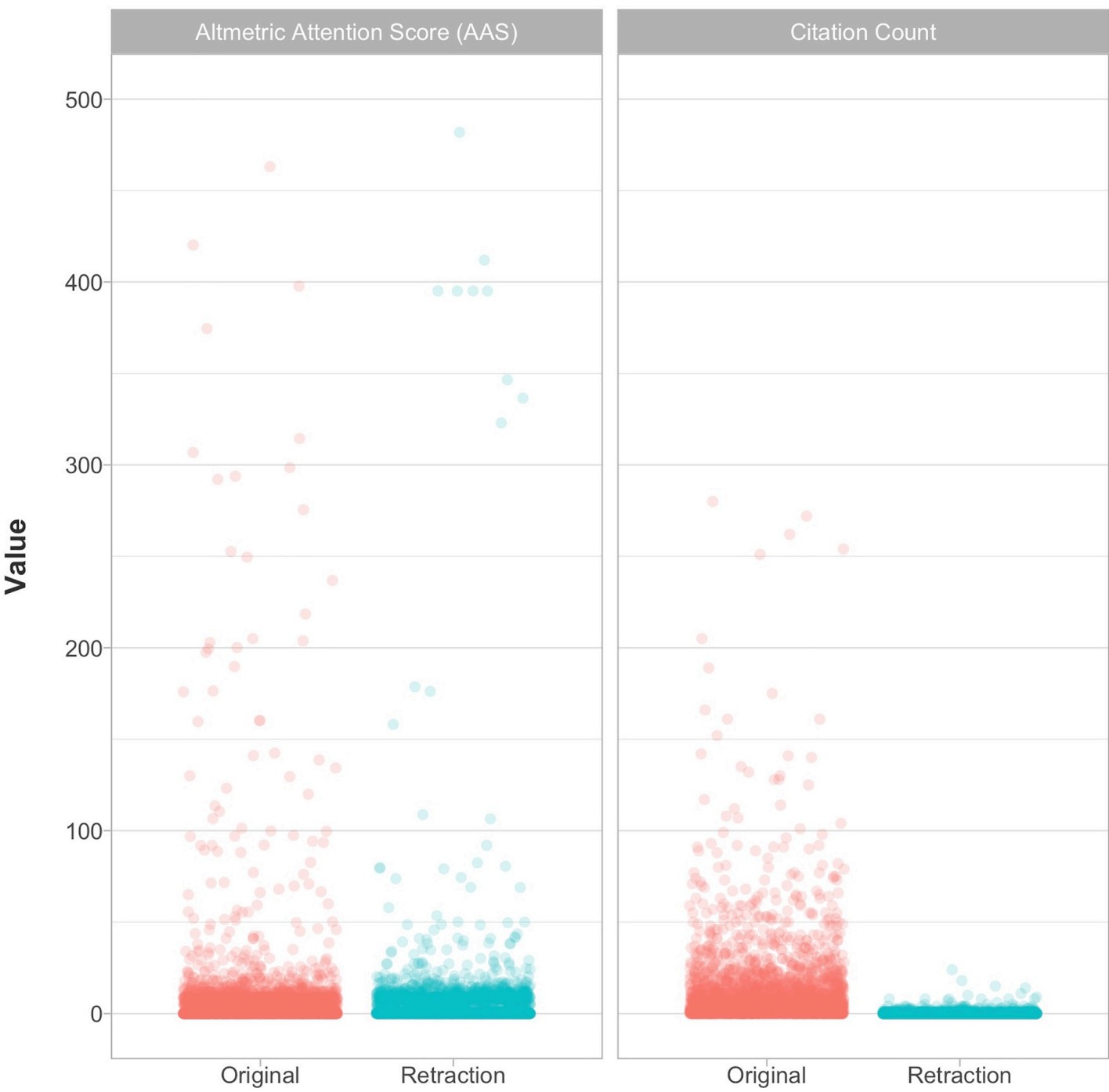

**Fig 2. AAS and citation count across original articles and retraction notices.** The distribution of AAS between 3,097 original articles and their retraction notices is fairly similar. However, a small number of original articles tend to receive more extreme AAS scores (5 articles with very high original AAS (Range, 1019–3166) are not shown for clarity). Unlike AAS, the citation count in 3,008 original articles is far greater than that in their retraction notices.

after the publication of the retraction) (S4 Table), but more substantially attenuated when this attention was then added to the retraction notice (S5 Table).

It could be that popular original articles attracted attention because of their retraction. As such, in a sensitivity study we examined the tweets associated with all 17/317 recently retracted

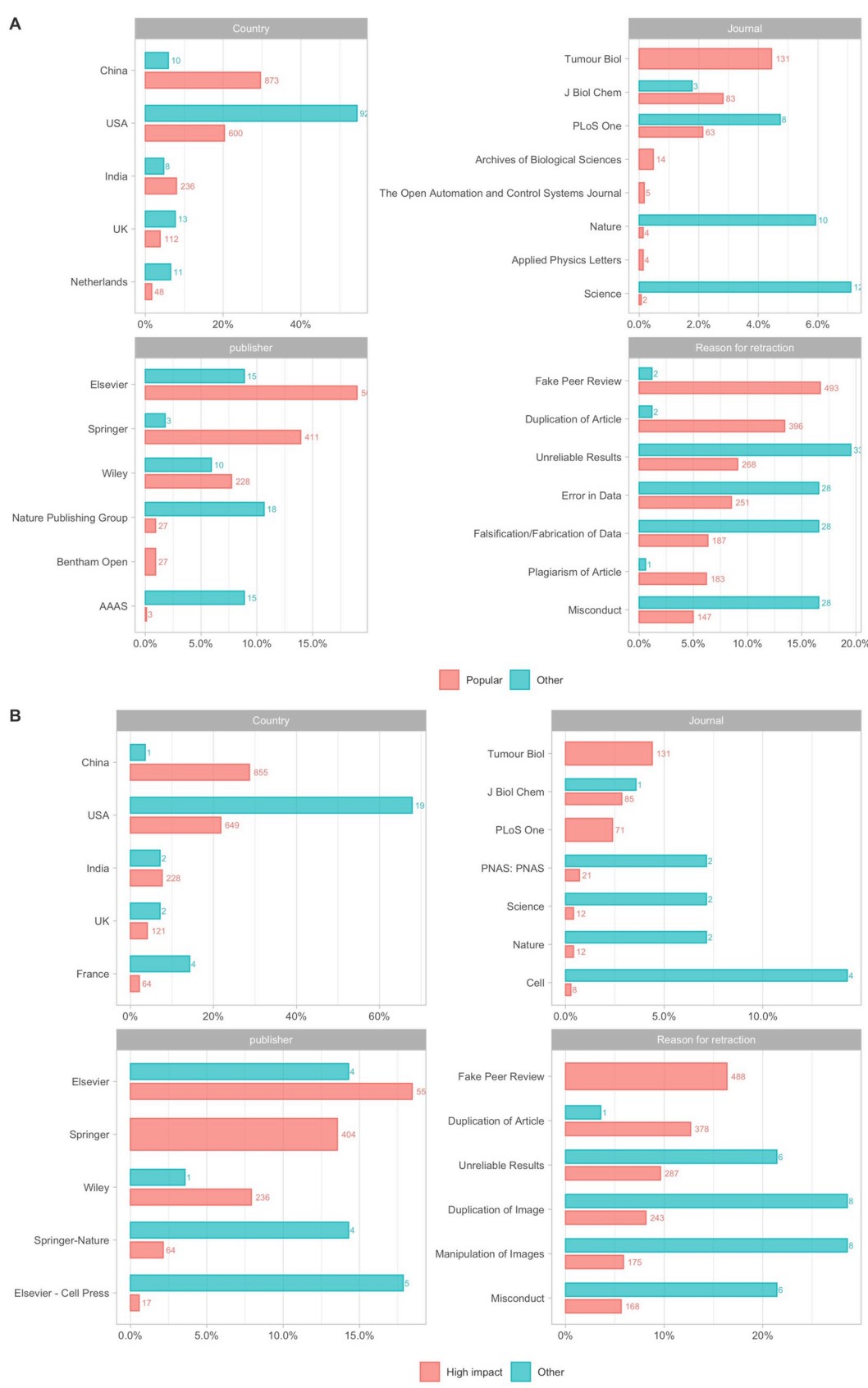

**Fig 3. Features across popularity and impact.** (A) The distribution of Country, Journal, Publisher and Reason for retraction is different across levels of popularity in 3,097 original articles. Popular retracted articles often came from the USA, were published in journals such as Nature and Science and were mostly retracted because of unreliable results. On the contrary, other retracted articles often came from China, were published in journals such as Tumor Biol and J Biol Chem and were primarily retracted because of fake peer review. (B) The distribution of Country, Journal, Publisher and Reason for retraction across levels of impact for 3,008 original articles had a similar pattern as the pattern seen across levels of Altmetric attention. However, the commonest reason for retraction in highly cited research (>100 citations) was duplication or manipulation of images.

articles that were popular even before retraction (317 instead of 279 because all original articles with a DOI were considered; also, note that tweets only constitute part of the AAS, which is a multi-factorial metric). 14/17 received at least one tweet (Median, 41; IQR, 12–95), of which 8/14 were openly available on Altmetric. Of these 8, the median number of pre-retraction tweets was 17 (IQR, 4–45) and none was negative. Similarly, the median number of post-retraction tweets was 2 (IQR, 1–3)—all tweets were negative for 5 articles, 1 article did not receive any tweets and, surprisingly, 2 articles exclusively received non-negative tweets [16, 17].

The first article was published in JAMA Pediatrics by Wansink et al. and concluded that branding school lunches can improve uptake of healthy food by school children [16]. This article received 4 non-negative tweets before retraction and 743 non-negative tweets after retraction—742/743 were retweets of a sentence that suggested that stickers make children choose fruit over cookies. The second article was published in the International Journal of Neuropsychopharmacology and concluded that ketamine is efficacious as a rapid-onset antidepressant in the emergency department [17]. It received 24 non-negative tweets before retraction and 2 non-negative tweets after retraction, both praising ketamine. In terms of citations, out of 3,008 records, the median difference in citations between the original and its retraction notice was 4 (IQR, 1–11; P-value $< 10^{-16}$) and for 31 highly cited original articles (>100 citations) it was 138 (IQR, 123–172; P-value $< 10^{-16}$). Overall, the original article received more citations than its retraction notice on 2,393 (80%) occasions and the retraction more than its original on only 122 (4%) occasions (for 493, the citations were equal).

## Attention to retracted vs. matched unretracted articles

We first compared 572 retracted articles matched with 2,832 unretracted articles, creating 450 distinct groups (see Methods). The largest such group contained 48 articles (40 unretracted, 8 retracted) and the smallest such group contained 3 articles (2 unretracted, 1 retracted) (Median, 6 articles; IQR, 6–6). Within groups, the median retracted article received a higher AAS than its median control on 253 occasions, a lower AAS on 57 occasions and the same AAS on 140 occasions (Relative risk, 4.43; 95% CI, 3.31–6.04). The median difference between median retracted and unretracted articles within groups was 0.50 (95% CI, 0.25–0.76).

We then restricted our analyses to articles retracted within a year of retrieving data from Altmetric (i.e. between June 6, 2017 and June 6, 2018). Out of 2,932 eligible articles with a unique PMID, 292 had been retracted within this time period. Of these, 55 had been matched with 387 unretracted articles, creating 47 distinct groups. The largest such group contained 31 articles and the smallest such group contained 4 (Median, 6 articles; IQR, 6–11). The large number of unretracted articles per group occurred because many groups originally contained more than 1 retracted article, which had been matched to 5 distinct unretracted articles.

Within matched groups, the median difference between retracted and unretracted articles in terms of all-time AAS was 1.00 (IQR, 0.00–7.35; 95% CI, 0.25–2.04) (Fig 6). After excluding the last year, the median difference between the two became 0 (IQR, 0–3.87; 95% CI, 0.00–0.00), despite the strong right skew (Mean, 6.06; 95% CI, 2.03–6.95). Out of 47 groups, in

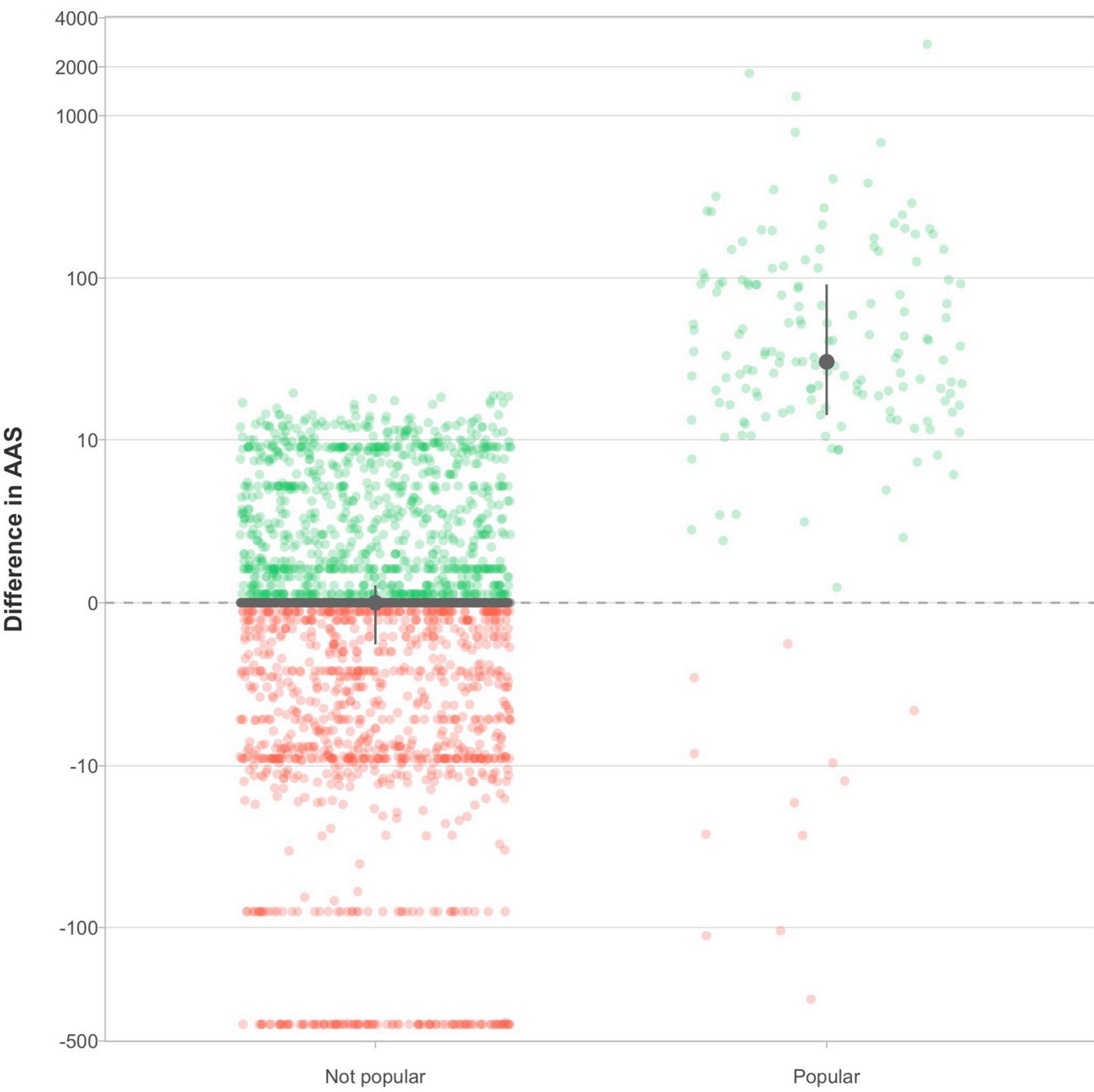

**Fig 4. Change in attention by popularity.** All points in green represent an original article that received more media and social media attention than its retraction notice and all points in red represent the opposite; points in grey represent no difference between the two. The large point and solid line in grey represent the median and its interquartile range. The difference is rather balanced for 2,923 articles that are not popular—the extreme negative values at -395 came from a single retraction in Tumor Biol, which retracted 103 unique original articles. In 168 popular articles, the difference is skewed to the right such that most popular articles did not have an equally popular retraction.

terms of all-time AAS, the median retracted article had a higher AAS on 30 occasions, versus 8 for the median unretracted article (P-value, 0.0005). Excluding the last year, the retracted article had a higher AAS on 23 occasions, versus 8 for the unretracted article (Relative risk, 2.88; 95% CI, 1.24–7.40; P-value, 0.01).

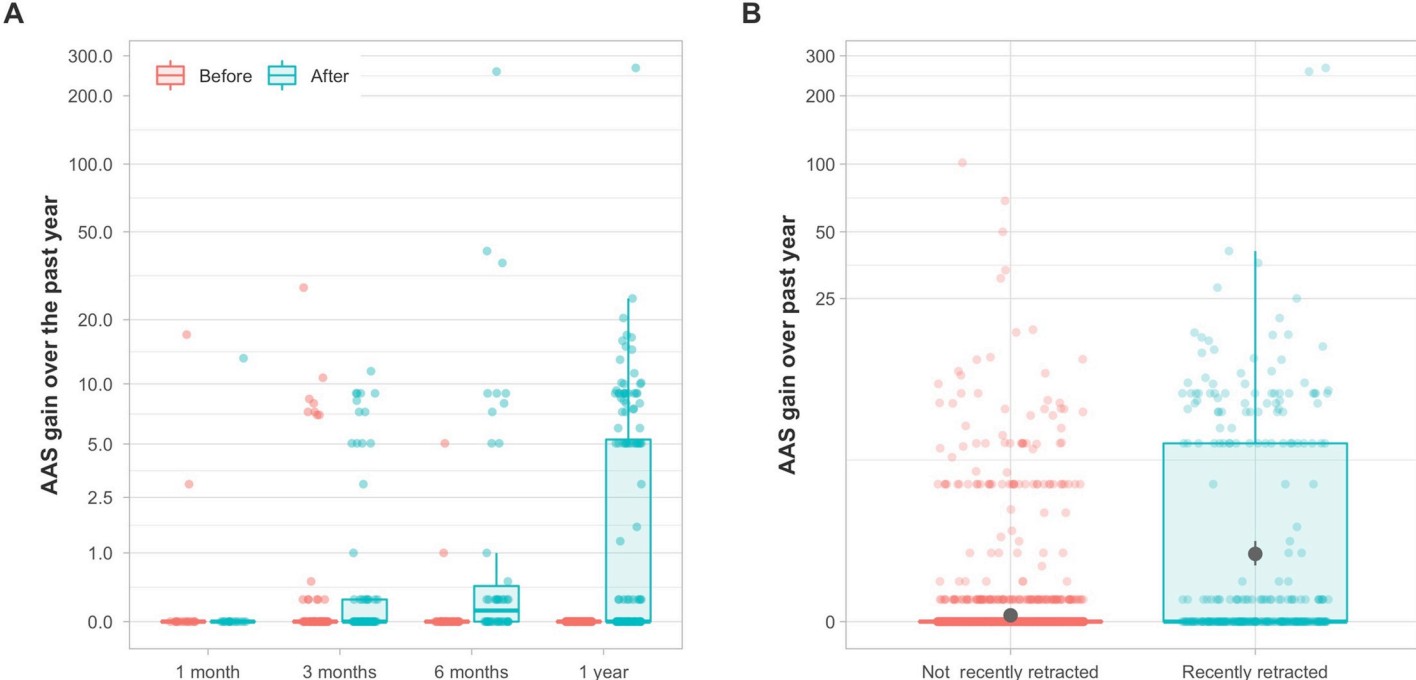

**Fig 5. Altmetric Attention Score (AAS) over time.** (A) Cumulative AAS for 279 articles retracted within a year of Altmetric data retrieval (from June 6, 2017 to June 6, 2018). The horizontal axis denotes how many months it has been since the article was retracted from the day of Altmetric data retrieval and the vertical axis the amount of AAS gained before and after retraction. It illustrates that, over the past year, most gains in AAS occurred after retraction. (B) Even though the median gain is zero in both 2,812 recently and 279 not recently retracted articles, proportionally many more recent articles experienced a gain, as denoted by the large interquartile range and the mean (grey dot denotes mean and range denotes the bootstrapped 95% CI).

## Discussion

In this literature-wide study of retraction, the number of retractions was found to vary substantially by country, journal, publisher and field of science. At least half of retracted research articles studied received some amount of Altmetric attention, almost 5% were considered popular and almost 1% had received more than 100 citations. Popular articles tended to receive substantially more attention than their retraction, even after adjusting for attention received post-retraction, and were often retracted because of unreliable data/results. This was unlike most other articles, where fake peer review was the most frequent reason for retraction. In

**Table 2. Pairwise comparison of original article vs. retraction notice with and without post-retraction AAS.**

|  | Overall | | Original ≥ 20 AAS | | Original > 0 AAS | |
|---|---|---|---|---|---|---|
|  | **Total** | **Before** | **Total** | **Before** | **Total** | **Before** |
|  | **N = 279** | **N = 275** | **N = 20** | **N = 15** | **N = 179** | **N = 124** |
| **Median difference (IQR)** | 0 (-0.3–1.5) | 0 (-1.0–0.3) | 31 (21–82) | 27 (15–66) | 0.8 (0–5) | 0.5 (-1–3) |
| **Median ratio (IQR)** | 1.4 (0.4–24.7) | 0.3 (0.0–4.3) | 4.1 (2.8–16.2) | 3.04 (2.3–11.4) | 2.5 (1.0–100.6) | 2.5 (0.5-Inf) |
| **Original > Retraction** | 121 (43%) | 79 (28%) | 20 (100%) | 14 (93%) | 121 (68%) | 79 (64%) |
| **Retraction > Original** | 82 (29%) | 123 (44%) | 0 (0%) | 1 (7%) | 42 (23%) | 40 (32%) |
| **Equal** | 76 (27%) | 77 (28%) | 0 (0%) | 0 | 16 (9%) | 5 (4%) |
| **P-value** | 0.007 | 0.002 | $2 \times 10^{-6}$ | 0.001 | $5 \times 10^{-10}$ | $4 \times 10{-4}$ |

Total = total AAS of original to date; Before = AAS received by original before retraction; Median ratio = median of the ratio AAS of original article / AAS of retraction notice; Inf = Infinity. The p-value is from a Binomial test for the number of articles with greater original vs. greater retraction attention.

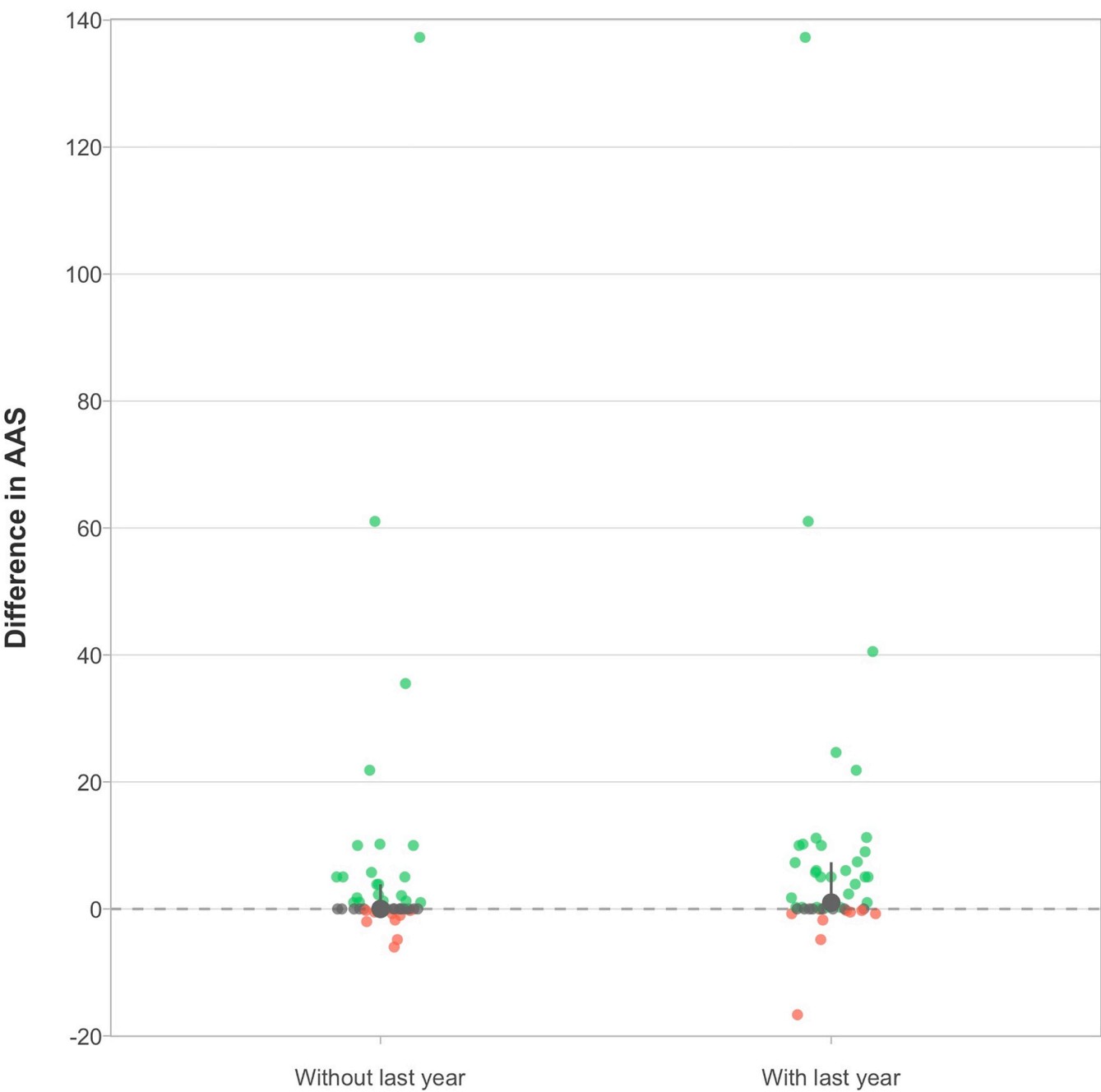

**Fig 6. Difference in attention within 47 matched groups of articles retracted within a year of Altmetric data extraction.** Each point represents the difference in AAS between the median retracted vs. the median unretracted article within matched groups. Green points represent groups in which the original article received more media and social media attention than matched unretracted articles and red points represent the opposite; points in grey represent no difference between the two. The "With last year" column represents all of the Altmetric attention received, whereas the "Without last year" column represents all Altmetric attention minus the last year. The large point and solid line in grey represent the median and its interquartile range. The median difference between retracted and matched unretracted articles is small (Median, 1) when including attention received over the last year and 0 otherwise. However, substantially more retracted articles received higher attention than matched unretracted articles, regardless of including attention from last year or not.

comparison to matched articles, retracted articles were 1.2–7.4 times more likely to receive more Altmetric attention, even after adjusting for attention received post-retraction.

Our results indicate that retracted articles do receive attention because of their original publication, but they also receive substantial attention because of their retraction. In fact, 100/175

(57%) retracted articles received most of their attention after retraction. However, this is not the case for the popular articles, which by the nature of being popular may also be the ones most likely to spread misinformation. These articles tend to receive 2.5 times the amount of attention received by their retraction after adjusting for attention received because of retraction.

There may be some reluctance of publishers to publish a retraction notice, to publish clear and informative notices and to make all potential readers of a retracted article aware that this article has been retracted [18]. Indeed, as we describe above, in one of our sensitivity analyses we surprisingly identified that in two of the articles for which post-retraction tweets were examined, retraction was directly or indirectly associated with the promotion of the initial misinformation (by being followed by further tweets spreading the initial results), rather than correcting the record. In a study of 88 articles by anaesthesiologist Dr. Boldt, which 18 journals had agreed to retract in 2011, 9/88 (10%) had yet to be retracted by 2013 [19]. Of the 79 retracted, only 15 (19%) were accompanied with an "adequate" retraction notice and only 48 (61%) were adequately marked as retracted. A similar study found that out of 235 studied retractions, 21 (9%) did not offer a detailed reason and 52 (22%) articles were available with no mention of retraction [20]. The problems are even worse for articles kept on central repositories or personal libraries [21, 22], despite clear guidelines by the Committee On Publication Ethics (COPE) [23] and the National Library of Medicine [24] recommendations.

All of these issues amalgamate into a critical problem, which is not inconsequential and which substantially hinders the ability of science to self-correct [25, 26]. In the presence of impediments, such as unclear and inconspicuous retractions, this self-correcting process may become unnecessarily slow, inefficient and ineffective [27].

The good news is that we do have the technology required to substantially improve our ability to flag retractions [28]. PubMed has been flagging retractions for years and it recently introduced a larger banner to help make retractions more apparent [29]. Similarly, the reference manager Zotero now automatically checks a user's database for retracted articles and issues a warning upon clicking on it and upon trying to cite it [30]. The effect of these initiatives on reducing misinformation remains to be studied.

## Limitations

This study has a number of limitations. First, this was a retrospective study, for which reason we could not access longitudinal data required to control our analyses for the possible effects of retraction over time. Even though we tried to adjust for the estimated effect of retraction by studying a subsample of our data with a suitable time window from retraction, more granular prospectively collected data would substantially help reduce risk of bias. Second, this was a descriptive, exploratory analysis. Even though we try to mitigate the bias inherent to exploratory analyses by presenting all of our analyses and avoiding focusing on p-values, a further pre-registered study would substantially reduce the risk of bias. Third, analyses of popular articles included a relatively small sample, especially when trying to adjust for the effects of retraction over the last year.

## Conclusions

Allowing for these limitations, our analysis documents that most eventually retracted articles and their retraction notice receive media and social media attention. In fact, eventually retracted articles tend to receive more media and social media attention than very similar, matched unretracted articles. Even though the original and its retraction tend to receive similar amounts of attention, popular articles receive substantially more attention than their

retraction notice. Such popular articles are most commonly retracted due to unreliable results, errors or misconduct, unlike other articles, which are primarily retracted due to fake peer review or duplication. Worryingly, popular articles receive additional attention upon retraction and this attention does not always reflect their retraction, but may perpetuate the original misconception.

## Materials and methods

This study uses a retrospective cohort design to investigate attention to the original articles vs. their retraction notice and a case-control design to investigate attention to retracted articles vs. matched articles that were not retracted. The report was compiled using the guiding principles of the STROBE statement for retrospective cohort studies [31].

### Attention to the original article vs. its retraction notice

**Data acquisition.** This is a retrospective cohort study of the retracted literature found in the Retraction Watch Database shared with us under a data use agreement on August 14, 2020 in compliance with their terms and conditions. The Retraction Watch Database is a repository of retracted articles and their retraction notice compiled by the Center of Scientific Integrity's Retraction Watch [32, 33] and represents the most comprehensive database of retracted literature that we know of. It was made available to the public in October, 2018 [2], at which point it hosted more than 18,000 articles published from the 1970s to 2018. Even though wherever applicable this manuscript describes the updated version of the database made available to us on August 14, 2020, all analyses utilizing Altmetric attention and citation data refer to the articles and retractions identified from a beta version of this database accessed on May 29, 2018.

We then downloaded all data available on Altmetric for the retrieved articles with a PubMed ID (PMID) or DOI on June 6, 2018 using the Altmetric Details Page API [13] and the R package rvest [34]. Altmetric gathers media and social media mentions of the published literature (e.g. mentions on Twitter, Facebook or news media), which it then compiles into a composite attention score, known as the Altmetric Attention Score (AAS) [35].

We retrieved citation data for all articles on the Retraction Watch Database with a PMID or DOI using the rcrossref package [36] in R. These are taken from Crossref, which is a not-for-profit association that interlinks and tracks citations between a variety of published research literature sources—at the time of writing Crossref had 46,723,946 articles with references deposited [37].

Finally, we extracted the total number of peer-reviewed documents published in science and engineering per country between 2010–2015 from the publication output table S5a-2 of the National Science Foundation [38]. We are using the total number of documents that mention each country at least once in their affiliations (called "whole count"), not the fraction of affiliations attributed to each country (called "fractional count").

**Eligibility criteria.** All peer-reviewed research articles on the Retraction Watch Database that were originally published between 2010–2015 and retracted by May 29, 2018 were eligible for our study. The 2010–2015 time-frame was chosen so as to allow sufficient time for most eventually retracted articles of this period to be retracted and for their retraction to receive most of the Altmetric attention it is likely to receive. We hereby define research articles as any studies labelled by Retraction Watch as any one of the following types of article: Research Article, Clinical Study, Meta-analysis, Letter. Preprints and dissertations were excluded because they are not peer-reviewed publications.

**Characteristic variables.** We retrieved and analyzed the following characteristic variables from the Retraction Watch database: title, author names, journal, publisher, institute, country

of affiliation, open access (yes or no), category (e.g. Physical sciences; determined by Retraction Watch), subcategory or subject (e.g. Geotechnical and Geological Engineering; determined by Retraction Watch), type of article (e.g. research article), type of notice (e.g. retraction, correction, etc.) and reason of retraction (e.g. plagiarism). The database also provided the following information for most original articles and related notices: PubMed ID (PMID), DOI and date of publication (for the original and its retraction).

**Outcome variables.**   The outcomes of interest were the total Altmetric Attention Score (AAS), the change in AAS across 1 month, 3 months, 6 months and 1 year since we last retrieved our data (these were the only time points made available by Altmetric) and the citation count. Altmetric indicates that articles with an AAS > 20 are thought to be doing "far better than their contemporaries" [39] - we call articles with an AAS > 20 in our sample "popular".

**Sensitivity analyses.**   Even though any attention received by the original article after its retraction was removed in comparing original articles to their retraction (for the subset of articles for which this information existed), it could be that original articles start accumulating retraction-related attention before their retraction. As such, we investigated the sensitivity of the observed effects to (a) removing all attention gained by popular articles over the past year, rather than only removing attention since their retraction and (b) adding the attention removed from the original article to the attention received by the retraction notice.

Likewise, it could be that some of the attention attributed to the original article emanates from concerns about the article (i.e. negative attention). To address this possibility, we examined all tweets gathered by Altmetric for all popular articles retracted within a year since we downloaded our Altmetric data. We then counted how many tweets were tweeted before and after retraction, how many of the pre- and post-retraction tweets we perceived based on our impression as negative (e.g. "this article has now been retracted") or non-negative (e.g. "this article presents impressive results") and copied the first negative and the first non-negative pre- and post-retraction tweet (always the first to avoid bias).

## Attention to retracted vs. matched unretracted literature

**Data acquisition.**   In addition to the aforementioned retrospective cohort study, we designed a case-control study. We treated all eligible articles retrieved from the Retraction Watch Database with a PMID as cases and then, for a random sample of 572 of these articles, we automatically identified a maximum of 5 random unretracted full articles from the same issue and journal. We indicate a "maximum of 5" as we could not always find 5 unretracted articles in the same issue. In the case of very large journal issues, such as issues that refer to a whole year, we matched cases to controls published for the first time (on PubMed this is "Date —Create") within the same issue and within 3 months of the publication of the case. We then extracted all information held by PubMed about the cases and matched controls using the package RISmed [40] in R. We also extracted all data from Altmetric and Crossref for the matched controls, as we had previously done for the cases.

**Eligibility criteria.**   All identified matched controls were eligible for further analysis.

**Variables.**   No characteristic variables other than those required to identify the matched groups were used in further analysis. The outcome variables for this study were the same as the ones for the retrospective cohort study described above.

## Missing and duplicated data

As far as we could tell, none of the characteristics or outcomes of interest were missing from the Retraction Watch Database. All records with a DOI or PMID and no Altmetric record we

assume have not received any attention, as indicated by Altmetric (personal communication). 287 original articles and 380 retraction notices with no DOI represent less than 10% of all 4,603 eligible unique records and thus they were excluded from Altmetric analysis with no attempt to impute missing data. We could not retrieve a citation count from CrossRef for 59/3,732 original articles with DOI and 71/3,647 retraction notices with a DOI—these also represent less than 10% of all eligible records, so all records with missing citation counts were excluded from the citation analysis with no attempt to impute the missing data.

The Retraction Watch database had two records for 11 articles with a DOI, each referring to a different notice. For example, an article may be retracted and replaced and then its replacement may also be retracted, leading to two separate retractions of an article with the same DOI.Even though it is possible to identify notices referring to the same original article by using the PMID or DOI, for 140/4,6011 (3%) records the database did not offer any of these two external and common identifiers. As such, for articles with no PMID or DOI, we identified notices referring to the same original article by using the title. Using this process we identified that none of the 140 articles was duplicated and we confirmed this by visual inspection.

## Statistical analysis

We produced descriptive statistics and visualizations for all exposure and outcome variables. Apart from date, all exposure variables were categorical and are presented in terms of counts and proportions. The outcome variables of interest (Altmetric Attention Score and citation counts) are heavily right-skewed, for which reason we primarily report the median and interquartile range, apart from a few exceptions where we felt the mean is also informative, in which case we present both. All continuous data were visualized in terms of boxplots and all discrete data in terms of bar charts. We also report the relevant sample size and missingness whenever applicable. All paired (original article and retraction notice) or grouped (retracted article and matched unretracted article) data were described both as grouped and ungrouped data and both in absolute (median difference of original vs. retraction) and relative (median ratio of original vs. retraction) terms.

Paired comparisons were done using the nonparametric Wilcoxon signed-rank test (for the comparison of the original article to its retraction notice), the Binomial sign test (for the comparison of proportions) and the non-parametric bootstrap with percentile confidence intervals (for the comparison of cases to matched controls). Results are presented in terms of effect size and the frequentist uncertainty in the effect size in terms of the p-value and the 95% Confidence Interval (CI).

To mitigate the potential bias inherent within exploratory analyses, such as this one, this report and the attached code contain the entirety of our analyses and all presented p-values and CIs were calculated and included after the completion of this report. All data processing and analysis was done in the programming language R [41].

## Data sharing

Data extracted from Altmetric, CrossRef and all matched PubMed articles have been deposited on OSF (Open Science Framework) and may be accessed at https://www.doi.org/10.17605/OSF.IO/7T32U under a CC-By Attribution 4.0 License. The Retraction Watch Database is available from Retraction Watch and requests for this data should be sent to: team@retraction-watch.com.

## Code sharing

The analytic code is available on GitHub under the GNU-3 License and may be accessed at https://github.com/serghiou/retraction-misinformation. We have also made all of our analyses available as a Markdown document, which can also be accessed on the same GitHub repository. We have turned the code required to download all data of interest from PubMed into an R package called metareadr available for download from https://github.com/serghiou/metareadr.

## Supporting information

**S1 Fig. Distribution of original publication versus retraction notice.** The distribution of retracted literature is roughly uniform across 6-month periods, whereas the distribution of their retractions follows a bell-curve with a left skew. This skew reflects that more publications are retracted very early rather than very late.
(TIF)

**S2 Fig. Altmetric attention before and after last year by recency of retraction.** There is no meaningful difference between the distribution of 2,733 articles retracted more than a year ago between AAS received before versus after the last year. On the contrary, for the 275 articles retracted within the last year, there is marked increase in total AAS when considering the last year. This implies that a substantial proportion of the total attention is received after retraction. The distribution of total AAS in articles that were recently retracted versus not, is not meaningfully different. The vertical axis is transformed to reflect $\log(AAS + 1)$. The grey dots are the mean and its bootstrapped 95% CI.
(TIF)

**S3 Fig. Change in attention by popularity in 275 recent research articles.** All points in green represent an original article that received more media and social media attention than its retraction notice and all points in red represent the opposite; points in grey represent no difference between the two. The large point and solid line in grey represent the median and its interquartile range. The difference is rather balanced for 260 articles that are not popular. In 15 popular articles, the difference is skewed to the right such that most popular articles did not have an equally popular retraction.
(TIF)

**S1 Table. Descriptive statistics for all 11,807 articles on Retraction Watch for 2010–2015.**
(HTML)

**S2 Table. Descriptive statistics for 4,603 research articles on Retraction Watch for 2010–2015.**
(HTML)

**S3 Table. Top 20 reasons for retraction in 4,603 research articles.** Count = number retracted for specific reason; Percent = proportion retracted for specific reason. Note that the proportions add up to >100% because articles could be retracted for more than one reason.
(DOCX)

**S4 Table. Pairwise comparison of original vs. retraction notice with or without counting the last year of Altmetric attention to the original.** The "Total" columns compare the total AAS received by the original articles against the total AAS received by their retraction notice. The "Without last" columns compare the AAS received by the original articles without the last year versus the total AAS received by their retraction notice. The values in parentheses are the

IQR for the median and the standard deviation for the mean. The p-value is from a Binomial test of articles with greater original vs. retraction attention. Not all articles for "Total" qualified for "Without last".
(DOCX)

**S5 Table. Pairwise comparison of original article vs. retraction notice when considering total post-retraction AAS.** The "Total" columns compare the total AAS received by the original article against the total AAS received by their retraction notice. The "Pre-retraction" columns compare the AAS received by the original article before retraction versus the attention received by its retraction notice plus any post-retraction attention directed to the original article. The values in parentheses are the IQR for the median and the standard deviation for the mean. The p-value is from a Binomial test of articles with greater original vs. retraction attention. Not all articles for "Total" qualified for "Pre-retraction".
(DOCX)

## Acknowledgments

We would like to thank Retraction Watch, Altmetric, Crossref and PubMed for making their data openly available to the scientific community. We would also like to thank the open source community of R developers and of ROpenSci, who provided many of the tools we used to complete this study. Finally, we thank our peer reviewers for substantial contributions.

## Author Contributions

**Conceptualization:** Stylianos Serghiou, Rebecca M. Marton, John P. A. Ioannidis.

**Data curation:** Stylianos Serghiou, Rebecca M. Marton.

**Formal analysis:** Stylianos Serghiou.

**Methodology:** Stylianos Serghiou, John P. A. Ioannidis.

**Supervision:** John P. A. Ioannidis.

**Visualization:** Stylianos Serghiou.

**Writing – original draft:** Stylianos Serghiou.

**Writing – review & editing:** Stylianos Serghiou, Rebecca M. Marton, John P. A. Ioannidis.

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
