## [Decision Letter · Decision Letter 0]

24 Jul 2020

PONE-D-20-17634

Media and social media attention to retracted articles according to Altmetric

PLOS ONE

Dear Dr. Ioannidis,

Thank you for submitting your manuscript to PLOS ONE. After careful consideration, we feel that it has merit but does not fully meet PLOS ONE’s publication criteria as it currently stands. Therefore, we invite you to submit a revised version of the manuscript that addresses the points raised during the review process.

We look forward to receiving your revised manuscript.

Kind regards,

Nikolaos Pandis

Academic Editor

PLOS ONE

Journal Requirements:

2. In your Methods section, please include additional information about your dataset and ensure that you have included a statement specifying whether the collection method complied with the terms and conditions for the websites from which you have collected data.

3.  Please indicate if the authors of the tweets included in Fig. 6 were contacted and agreed for their tweets to be reproduced in this paper. Alternatively, please confirm with your IRB, data protection committee or other relevant research ethics committee that consent from these authors is not required.

Reviewers' comments:

Reviewer's Responses to Questions

**Comments to the Author**

1. Is the manuscript technically sound, and do the data support the conclusions?

Reviewer #1: Partly

Reviewer #2: Yes

2. Has the statistical analysis been performed appropriately and rigorously? 

Reviewer #1: Yes

Reviewer #2: Yes

3. Have the authors made all data underlying the findings in their manuscript fully available?

Reviewer #1: Yes

Reviewer #2: Yes

4. Is the manuscript presented in an intelligible fashion and written in standard English?

Reviewer #1: Yes

Reviewer #2: Yes

5. Review Comments to the Author

Reviewer #1: Thank you for the opportunity to review this manuscript. Overall, this approach to the topic of retractions is very timely and innovative.

However, as we detail below, the authors appear to have relied on a dataset that was clearly marked “beta” at the time they extracted the data (May 2018) and was not launched until October 2018. At the very least, that fact should be clearly referenced throughout the manuscript (and the supplementary information, publication of which violates the data use agreement scholars have agreed to in exchange for the data). But more to the point, that seems to throw into doubt the accuracy and reproducibility of the findings. We would suggest that the authors contact us -- the creators of the Retraction Watch Database -- for a complete download of the dataset, at which point they can repeat their analyses with confidence.

Specific:

Introduction

Page 4: “Retraction refers to the formal withdrawal of a peer-reviewed publication, mostly due to …” the referenced citation does not have a stipulation for “peer-reviewed, and to be accurate the “mostly due to” should probably read “most often due to”. (The reference says: “This is the formal withdrawal of one or more papers by one or all of the authors. In most circumstances, retraction happens when new findings, or an inability by other groups to replicate results, spur the authors to withdraw a paper.”)

Page 4: “These results align well with broader studies,” The three citations refer to specific cases or a medical specialty – which do not count as “broader studies.” Broader studies to cite might be: https://asistdl.onlinelibrary.wiley.com/doi/full/10.1002/pra2.2016.14505301055 (disclosure: one of the reviewers is a co-author), or https://asistdl.onlinelibrary.wiley.com/doi/10.1002/pra2.35. There are others.

Page 5: “authors of guidelines and meta-analyses seem to very rarely update their work to remove retracted articles [10,11]” Citation 10 is a review (not research) article – it would be a secondary source at best, but it does not discuss guidelines or meta-analyses. In addition, while an author may request revisions, corrections or otherwise, a journal/publisher may refuse to allow them, so perhaps taking “authors” out of the equation and simply saying these are very rarely updated would be more accurate.

Page 5: “We hereby integrated the Retraction Watch Database..” Minor point, but “hereby” seems needlessly formal and can be deleted.

Results

Page 6: “Out of 4,217 research articles, 3,972 (94.2%) received a retraction notice (Table 1; S2 Table). This needs some context, and we would refer the authors to question 11 of the Retraction Watch Database User Guide: https://retractionwatch.com/retraction-watch-database-user-guide/ Corrections and Expressions of Concern (EoCs) have been entered as a point of interest related to authors with retractions. Unlike with retractions, there is no intention to be comprehensive on other types of citations.

Page 8: “It is unclear whether higher rates of retraction in certain journals/countries are due to less robust research practices versus more robust efforts to identify papers that should be retracted or confounded by other factors (e.g. type and impact of research done)” This comment seems more appropriate in a discussion section, and would be improved with some citations or evidence to support the speculation.

Page 11: “The difference is rather balanced for 2,843 articles that are not popular - the extreme negative values around -400 came from a single retraction in Tumor Biol.” Was this the retraction notice that pulled more than 100 articles at one time? This would make the weighting of this notice very unequal to other “single article” notices.

Page 14: “2 articles only received non-negative tweets [16,17] .” Citation 16 and 17 belong to the articles presumably receiving “non-negative tweets”. Perhaps the authors could change it to “2 articles only (Wansink et al. and Larkin et al.) only received non-negative tweets [16,17]” to make it clear that the citations do not belong to a substantiation of the tweet numbers. The authors identify the two in the subsequent paragraph – but perhaps best to clarify it in the first sentence.

Page 14: “Green signifies a non-negative tweet and red signifies a negative tweet.” Using colors as the sole indicator is problematic in readability for colorblind readers, which one of the reviewers is. Could the authors combine the colors with some other type of indicator – such as bolded letter vs non-bolded, stars and squares for data points instead of simply colors?

Discussion:

Page 18: “At least half of retracted research articles received some amount of Altmetric attention.” Would suggest adding “studied” after “articles” to clarify the sample analyzed.

Page 19: “Indeed, in one of our sensitivity analyses we surprisingly noticed that in two of the articles examined, retraction led to the promotion of the initial misinformation (by being followed by further tweets spreading the initial results), rather than correcting the record.” We would suggest adding an example to demonstrate this.

Page 19: “PubMed has been flagging retractions for years and it recently introduced a larger banner to help make retractions more apparent [28].” The authors may want to note that the banner effect is only available when the journal/publisher sends the retraction information in an appropriate format so the articles are linked. Retractions are still showing up without the original article being flagged, based on the metadata and information provided by the journal/publisher.

Page 20: “Finally, a number of inconsistencies were identified while working with the data extracted from the Retraction Watch Database (see Methods). Even though we tried to correct as many of these as we could, it is possible that more of these exist.” It is deeply problematic to include this comment without noting that the authors chose to scrape a database clearly marked “beta” at the time, and did not contact the (well-known) creators of the database to either request a full download -- which is made freely available to scholars subject to a simple data use agreement -- or query the inconsistencies. The “beta” designation was not removed until October 2018, when the database was officially launched: https://www.sciencemag.org/news/2018/10/what-massive-database-retracted-papers-reveals-about-science-publishing-s-death-penalty As noted in our general comments, we would strongly recommend that the authors now request a download and repeat their analyses. If they choose not to do so, the manuscript requires prominent notices that the authors chose to use data marked “beta” that was known to be incomplete at the time.

Methods:

Page 22: “It was made available to the public in October, 2018 [2] … We extracted all records available on this database as of May 29, 2018 that were originally published between 2010-2015.” This sentence is somewhat confusing. It suggests, as we have noted elsewhere, that the authors extracted all the data prior to the “beta” designation being removed. If so, how did they extract it? Or did they extract it after the database went public, in which case it is unclear how they would know what was available on May 29, 2018? Did they have permission to acquire the information, as Retraction Watch has data usage requirements?

If the data were extracted prior to the public data, then they were not complete, in which case the data and thus the analyses/results are based on incomplete and potentially biased data without being marked that way. While no resource will ever be 100% comprehensive for a variety of reasons that make retraction notices very difficult to find (e.g., notice in print form only and yet to be discovered, foreign language issues, journals removing articles from Table of Contents without notices, etc.), the authors have exacerbated these issues without noting that limitation in their manuscript, instead referring without comment to inconsistencies.

Also, cross-checking the database for retractions made up to May 29, 2018, using Article type: Research Article or Clinical Study or Letter, and restricting the dates of publication from 01/01/2010 to 12/31/2015, only 4108 entries are returned. Again, the data until this point were incomplete, and the authors extracted data before a “study” was completed and are using the data prematurely.

Page 23: “A publication was considered a research article when labelled by Retraction Watch Database as any of: Research Article, Clinical Study, Clinical Study/Research Article, Letter or Letter/Research Article.” The number of retractions was compared with the number of publications per year, using the NSF counts (ref 36), but the NSF excludes letters in their counts: “The articles exclude editorials, errata, letters, and other material that do not present or discuss scientific data, theories, methods, apparatuses, or experiments. The articles also exclude working papers, which are not generally peer reviewed.” Did the authors confirm that the retractions associated with “Letters” would have been included in the NSF counts?

Page 23: “In 37/3,972 research articles, the reported date for the original article was paradoxically later than the reported date for its retraction - we excluded these articles when reporting on the duration between publication of the original and its retraction.” Why did the authors not just attempt to locate the correct dates and then include the articles? Did they consider contacting the creators of the database to alert them to the pre-launch errors?

Page 23: “Altmetric indicates that articles with an AAS > 20 are thought to roughly represent the top 5% of the literature” Citation?

Page 26: “we automatically identified a maximum of 5 random unretracted full articles from the same issue and journal. We indicate a “maximum of 5” as we could not always find 5 unretracted articles in the same issue.” The concept of randomness is fine - except when the selections for control are so disparate. Perhaps the authors could separate journal issues into groups of comparable sizes, then choose random samples from each.

Page 25: “we examined all tweets gathered by Altmetric for all popular articles retracted within a year since we downloaded our Altmetric data.” As Altmetric uses multiple sources (blogs, tweets, various main and social media sources), using tweets as an indication of the positive or negative measure of attention seems a bit narrow. Users of Twitter is a very biased group in itself (e.g. https://www.pewresearch.org/internet/2019/04/24/sizing-up-twitter-users/), may actually not be the widest used social media commenting platform, (https://blog.hootsuite.com/twitter-demographics/), and may not be appropriately comparable to the measure of attention an article is receiving across main media outlets or filed-related blogs. Did the authors cross-reference their Twitter findings against other platforms and genres?

Page 27: “for 2,944/10,370 (14.5%) records the database did not offer any of the two and did not offer any other unique identifier” The database does not issue these numbers -- PubMed and CrossRef do, when publishers apply for them -- and would only have them if they were available. Perhaps language change to “No PMID or DOI were available for 2,944/10,370 (14.5%) records in the database, which relies on these two external and common identifiers.”

Reviewer #2: This was an interesting work examining the Altmetric Attention Score for retracted papers both before and after retraction. The authors chose 2010-2015 works from RetractionWatch and grabbed data from Crossref, PubMed and Altmetric.com to supplement information. My small concern that the authors should address is that Altmetric.com started tracking attention Oct, 2011. Thus, articles published before Oct 2011 could have misleading AAS and this needs to be addressed in the paper.

The statistics, supplemental information, and overall stats seem appropriate, but I am not an expert on statistics and would defer to other peer reviewers with more knowledge in this area to determine if other measures could/should have been used.

Overall, I found the work to be well written and logical. The information visualizations seem appropriate and paint the picture the authors are telling.

I would consider focusing a bit more on how positive/negative tweets were analyzed; was there a sentiment analysis performed and how was it performed? Was it just based on authors' impressions? More clarity here, please. This is of interest because it suggests that Twitter users were aware of the retraction and posted messages to help others become aware. Were these tweets warnings to others? Did they call out misinformation spread? There are plenty of articles performing sentiment analysis on tweets and the authors should perform a quick review and specify what is applicable to tweets and what is not (e.g., http://ucrel.lancs.ac.uk/crs/attachments/UCRELCRS-2013-05-16-Thelwall-Slides.pdf).

6. PLOS authors have the option to publish the peer review history of their article (what does this mean?). If published, this will include your full peer review and any attached files.

Reviewer #1: **Yes: **Ivan Oransky and Alison Abritis

Reviewer #2: No

---

## [Author Response · Author response to Decision Letter 0]

23 Jan 2021

We sincerely thank the reviewers for thorough and insightful feedback. All of our responses to their comments may be found within the "Response to reviewers" document.

---

## [Decision Letter · Decision Letter 1]

16 Feb 2021

PONE-D-20-17634R1

Media and social media attention to retracted articles according to Altmetric

PLOS ONE

Dear Dr. Ioannidis,

Thank you for submitting your manuscript to PLOS ONE. After careful consideration, we feel that it has merit but does not fully meet PLOS ONE’s publication criteria as it currently stands. Therefore, we invite you to submit a revised version of the manuscript that addresses the points raised during the review process.

We look forward to receiving your revised manuscript.

Kind regards,

Nikolaos Pandis

Academic Editor

PLOS ONE

Reviewers' comments:

Reviewer's Responses to Questions

**Comments to the Author**

1. If the authors have adequately addressed your comments raised in a previous round of review and you feel that this manuscript is now acceptable for publication, you may indicate that here to bypass the “Comments to the Author” section, enter your conflict of interest statement in the “Confidential to Editor” section, and submit your "Accept" recommendation.

Reviewer #1: (No Response)

2. Is the manuscript technically sound, and do the data support the conclusions?

Reviewer #1: Yes

3. Has the statistical analysis been performed appropriately and rigorously? 

Reviewer #1: Yes

4. Have the authors made all data underlying the findings in their manuscript fully available?

Reviewer #1: No

5. Is the manuscript presented in an intelligible fashion and written in standard English?

Reviewer #1: Yes

6. Review Comments to the Author

Reviewer #1: We appreciate the authors' responsiveness to previous comments. We only have a few minor comments to add:

1. In Methods, the authors note: “we extracted the total number of peer-reviewed documents”, and in the following paragraph “peer-reviewed” was added to the statement: “All research articles on the Retraction Watch Database that were originally published between 2010-2015...” The authors further added: “Preprints and dissertations were not included because they are not peer-reviewed publications.” Did the authors confirm that all the articles in their sample were indeed from peer-reviewed journals as there is no filtering requirement for peer-reviewed journals for inclusion in the database?

2. We previously commented about a paragraph from Page 5 in the prior manuscript: “Page 5: “authors of guidelines and meta-analyses seem to very rarely update their work to remove retracted articles [10,11]” Citation 10 is a review (not research) article – it would be a secondary source at best, but it does not discuss guidelines or meta-analyses. In addition, while an author may request revisions, corrections or otherwise, a journal/publisher may refuse to allow them, so perhaps taking “authors” out of the equation and simply saying these are very rarely updated would be more accurate.” There are no author comments regarding this, although the citation now is changed to “11,12.” Again, citation (now) 11 is a review article and does not discuss guidelines and meta-analyses in particular. And we reiterate that authors may have little control over the updating of their work. So again – perhaps modifying the reference choice and changing the language to remove the perceived burden of updating being solely that of the authors would be more appropriate.

3. The data availability section for the manuscript acceptance says ”yes- all data are fully available without restriction.” However, elsewhere the authors note (correctly) that data from the Retraction Watch Database are available from the Center For Scientific Integrity. We would suggest ensuring that these responses are consistent.

7. PLOS authors have the option to publish the peer review history of their article (what does this mean?). If published, this will include your full peer review and any attached files.

Reviewer #1: **Yes: **Ivan Oransky and Alison Abritis

---

## [Author Response · Author response to Decision Letter 1]

28 Feb 2021

All responses have been included in the "Response to Reviewers" document.

---

## [Decision Letter · Decision Letter 2]

5 Apr 2021

Media and social media attention to retracted articles according to Altmetric

PONE-D-20-17634R2

Dear Dr. Ioannidis,

We’re pleased to inform you that your manuscript has been judged scientifically suitable for publication and will be formally accepted for publication once it meets all outstanding technical requirements.

Kind regards,

Pablo Dorta-González, Ph.D.

Academic Editor

PLOS ONE

Additional Editor Comments (optional):

Reviewers' comments:

Reviewer's Responses to Questions

**Comments to the Author**

1. If the authors have adequately addressed your comments raised in a previous round of review and you feel that this manuscript is now acceptable for publication, you may indicate that here to bypass the “Comments to the Author” section, enter your conflict of interest statement in the “Confidential to Editor” section, and submit your "Accept" recommendation.

Reviewer #1: All comments have been addressed

2. Is the manuscript technically sound, and do the data support the conclusions?

Reviewer #1: Yes

3. Has the statistical analysis been performed appropriately and rigorously? 

Reviewer #1: Yes

4. Have the authors made all data underlying the findings in their manuscript fully available?

Reviewer #1: No

5. Is the manuscript presented in an intelligible fashion and written in standard English?

Reviewer #1: Yes

6. Review Comments to the Author

Reviewer #1: We thank the authors for responding thoughtfully and comprehensively to our comments, and we recommend acceptance.

7. PLOS authors have the option to publish the peer review history of their article (what does this mean?). If published, this will include your full peer review and any attached files.

Reviewer #1: **Yes: **Ivan Oransky and Alison Abritis

---

## [Editor Report · Acceptance letter]

9 Mar 2021

PONE-D-20-17634R2 

Media and social media attention to retracted articles according to Altmetric 

Dear Dr. Ioannidis:

I'm pleased to inform you that your manuscript has been deemed suitable for publication in PLOS ONE. Congratulations! Your manuscript is now with our production department. 

Kind regards, 

on behalf of

Dr. Nikolaos Pandis 

Academic Editor

PLOS ONE